# Design of an Integrated Remote and Ground Sensing Monitor System for Assessing Farmland Quality

**DOI:** 10.3390/s20020336

**Published:** 2020-01-07

**Authors:** Feiyang Zhang, Guangxing Wang, Yueming Hu, Liancheng Chen, A-xing Zhu

**Affiliations:** 1College of Resources and Environment, South China Agricultural University, Guangzhou 510642, China; eeefy@163.com; 2Guangdong Provincial Key Laboratory of Land Use and Consolidation, South China Agricultural University, Guangzhou 510642, China; 3Guangdong Province Engineering Research Center for Land Information Technology, South China Agricultural University, Guangzhou 510642, China; 4Key Laboratory of Construction Land Transformation, Ministry of Land and Resources, South China Agricultural University, Guangzhou 510642, China; 5Department of Geography and Environmental Resources, Southern Illinois University, Carbondale, IL 62901, USA; 6Department of Geography, University of Wisconsin Madison, Madison, WI 53706, USA

**Keywords:** wireless sensor network, unmanned aerial vehicles, scheduling algorithms, integrated design

## Abstract

Quality monitoring is important for farmland protection. Here, high-resolution remote sensing data obtained by unmanned aerial vehicles (UAVs) and long-term ground sensing data, obtained by wireless sensor networks (WSNs), are uniquely suited for assessing spatial and temporal changes in farmland quality. However, existing UAV-WSN systems are unable to fully integrate the data obtained from these two monitoring systems. This work addresses this problem by designing an improved UAV-WSN monitoring system that can collect both high-resolution UAV images and long-term WSN data during a single-flight mission. This is facilitated by a newly proposed data transmission optimization routing protocol (DTORP) that selects the communication node within a cluster of the WSN to maximize the quantity of data that can be efficiently transmitted, additionally combining individual scheduling algorithms and routing algorithms appropriate for three different distance scales to reduce the energy consumption incurred during data transmission between the nodes in a cluster. The performance of the proposed system is evaluated based on Monte Carlo simulations by comparisons with that obtained by a conventional system using the low-energy adaptive clustering hierarchy (LEACH) protocol. The results demonstrate that the proposed system provides a greater total volume of transmitted data, greater energy utilization efficiency, and a larger maximum revisit period than the conventional system. This implies that the proposed UAV-WSN monitoring system offers better overall performance and enhanced potential for conducting long-term farmland quality data collection over large areas in comparison to existing systems.

## 1. Introduction

Farmland is a basic resource required for ensuring human survival. However, the quality of farmland dictates the quantity and quality of the extracted crops, which, in turn, affect the quality and health of human life [1,2]. As such, farmland quality monitoring is a key task required for farmland protection [3]. Presently, farmland quality monitoring is predominantly conducted via laboratory analyses. However, the development of science and technology has provided new monitoring methods uniquely suited for assessing temporal and spatial changes in farmland quality. Among these, high-resolution remote sensing data obtained by unmanned aerial vehicles (UAVs) and long-term ground sensing data obtained by wireless sensor networks (WSNs), like soil moisture data obtained from three different depths, meteorological data and so on, are two key monitoring methods with substantial potential for conducting farmland quality monitoring [4,5,6,7]. Moreover, the respective spatial and temporal benefits provided by these separate methods has generated considerable interest in the integration of the data collected by these two types of monitoring systems into a single farmland quality monitoring system [8,9,10,11,12,13,14].

One key component required for effectively integrating the data collected by UAVs and WSNs is the development of a suitable method for transmitting the data collected by a WSN to the UAV as it passes over the monitored region. Currently, three data transmission methods have been developed for this purpose, which include point-to-point transmission, the flat routing protocol, and the clustering routing protocol.

Point-to-point data transmission utilizes a static star topology, where the sensor nodes of the WSN located on the ground do not communicate with each other, but rather send their data to a sink node in a passing UAV directly [15,16,17,18,19,20,21,22]. This data transmission method adopts a simple network structure that is easy to build. However, the UAV must traverse every sensor node in the WSN to collect all the available data, which results in an irregular UAV flight route that increases flight time and consumes a greater proportion of available battery energy. Moreover, the irregular flight route can result in the wasteful overlap of remote sensing images, which further reduces the monitoring efficiency. Accordingly, point-to-point data transmission is only suitable for use under conditions where the density of sensor nodes is low.

The flat routing protocol is a dynamic routing protocol in which all sensor nodes in the WSN are treated as equivalent and serve the same function [23]. Here, sensor nodes send their data to route nodes at fixed locations according to a number of possible flat routing protocols, such as directed diffusion (DD) or sensor protocol for information via negotiation (SPIN), and then only the route nodes send their data to the sink node in the passing UAV directly [24,25,26]. This method has been employed to good effect in a number of UAV-WSN applications. For example, João et al. [27] adopted the flat routing protocol in a UAV-WSN system designed to monitor isolated vineyards separated by substantial distances. Here, several sensor nodes and a single route node were deployed in each vineyard based on the flat routing protocol. Hence, the UAV was required to fly over the route nodes of the isolated vineyards only. While this method can greatly streamline the flight path of a UAV, and thereby reduce its flight time and energy consumption, the sensor nodes of the WSN suffer from unbalanced energy consumption during data transmission, owing to their varying distances from the single route node. Moreover, the streamlined flight route of the UAV over the route nodes only reduces the coverage of UAV imaging. Accordingly, the flat routing protocol is most suitable for conditions where the sensor nodes are separable into several groups.

The clustering routing protocol is another dynamic network protocol where variable clusters of sensor nodes only communicate with specifically selected head nodes [28]. Here, only the head nodes of the presently configured clusters directly send their data to the sink node in a passing UAV, and the changing position of the UAV is accommodated by reconfiguring the individual clusters of the sensor nodes in the WSN and reselecting the respective head nodes after each data transmission in order to avoid overtaxing the battery resources of the selected head nodes [29,30,31,32,33]. Research focused on the development of UAV-WSN systems based on the clustering routing protocol has featured a number of methods for configuring sensor node clusters, such as particle swarm optimization (POS) or the weighted K-means clustering algorithm [34,35,36,37,38], and the head nodes are selected by other methods, such as the low-energy adaptive clustering hierarchy (LEACH) protocol or the hybrid energy-efficient distributed (HEED) clustering approach [39,40,41]. Finally, the optimal flight route of the UAV over the head nodes is typically planned using a number of approaches, such as the traveling salesman problem (TSP) or simulated annealing (SA) [29,30,42,43,44,45,46]. This method has also been employed to good effect in a number of UAV-WSN applications. For example, Huiru et al. [37] deployed a UAV-WSN system based on the clustering routing protocol over a large area for the purpose of environmental monitoring. The weighted K-means algorithm was employed for sensor node clustering, the LEACH protocol was employed to select the head node of each cluster, and SA was employed to plan the flight route. According to its operational mode, the clustering routing protocol has a strong capability for balancing the energy consumption of sensor nodes in a WSN. However, like the flat routing protocol, the streamlined flight route of the UAV, flying over the head nodes only, reduces the coverage of UAV imaging. Moreover, the changing head node locations of the clustering routing protocol induces an additional disadvantage, in that the images collected during different flight missions cannot be readily compared due the corresponding changes in the flight route. Therefore, the clustering routing protocol is best suited for conditions where the monitored area and the density of sensor nodes are both large.

The common disadvantage of the data transmission protocols discussed above for use in UAV-WSN system applications is that they do not provide sufficient flexibility to UAVs for the collection of remote sensing data due to the constraints associated with the collection of ground sensor data [47,48,49,50]. Presently, this necessitates that the UAV must fly over a study area twice, where one flight path is planned for visiting each communication node in the WSN (i.e., the sensor nodes, route nodes, or head nodes, depending on the protocol in use) and another is planned for collecting high-resolution remote sensing data. However, the combined collection of both ground sensor and remote sensing data over a single-flight path by an integrated UAV-WSN system would greatly reduce the flight time and energy usage of UAVs during farmland quality monitoring operations, while also enabling the simultaneous collection of high-resolution remote sensing data. Therefore, the facile and flexible integration of ground sensor and high-resolution remote sensing data in a UAV-WSN system requires the development of a new data transmission protocol specifically designed for this purpose. As such, the designed protocol must accommodate the flexible routing of UAVs and adhere to the principles of aerial photogrammetry while enabling the efficient collection of ground sensor data.

These issues are addressed in this work by developing an integrated UAV-WSN system based on a newly proposed data transmission optimization routing protocol (DTORP). In the following, Section 2 presents the design of the improved UAV-WSN system. Section 3 presents the DTORP. Section 4 applies Monte Carlo simulations to analyze the performance of our proposed system in comparison with the performance obtained using a conventional system based on the LEACH protocol. Finally, conclusions are presented in Section 5.

## 2. Improved UAV-WSN System Design

Like most existing UAV-WSN systems, the WSN of the improved UAV-WSN system proposed in this study is composed of sensor modules, communication modules, control modules, and power supply modules. Similarly, the UAV is an unmanned drone craft, carrying a camera or imaging spectrometer to capture images or spectral data for the study area and a sink node for collecting ground sensor data from the WSN.

As discussed, the working process of most existing UAV-WSN systems is to plan the flight path of the UAV according to the positions of communication nodes in the WSN. However, the working process of our proposed UAV-WSN system facilitates the simultaneous collection of ground sensor and high-resolution remote sensing data by planning the UAV flight path independent of the WSN, according to the principles of aerial photogrammetry, which ensures that the images or spectral data can be used to produce an aerial view using standard remote sensing products, such as digital surface models. Then, sensor node clustering is conducted for the WSN according to the selected UAV flight path, and every sensor node is considered a potential communication node to ensure that the UAV can connect to any node of a sensor node cluster as the UAV passes over a given region of the study area from any direction. The individual nodes in the sensor node clusters are connected with the sink node of the UAV by a newly proposed fast broadcast routing protocol (FBRP) that facilitates the collection of all available sensor node data by the UAV. Finally, the UAV applies the proposed DTORP for conducting data transmission within each sensor node cluster.

The FBRP is similar to a flooding protocol in principle. The sink node in the UAV continuously broadcasts a group order. The first sensor node in a cluster to receive the order from the sink node in the UAV broadcasts the order to its neighboring sensor nodes. Then, each of the neighboring sensor nodes broadcasts the order to their neighboring sensor nodes. This process continues until each sensor node in the cluster has received the order. Once a sensor node receives the order from a neighboring sensor node, it will ignore the order later transmitted from any other neighboring node. After receiving the order, each sensor node transmits its data quantity, battery status, and neighboring nodes to the sink node in the UAV along the same route by which the order arrived from the sink node to the sensor node. In contrast to most existing data transmission protocols, which are designed to reduce energy consumption or balance the consumption of battery energy, the proposed protocol is designed to reduce the overall time required for collecting all available sensor node data.

## 3. Data Transmission Optimization Routing Protocol

The DTORP was developed to increase the volume of data transmission and decrease and balance battery consumption in the improved UAV-WSN system. The working principle of the DTORP consists of two primary components, namely, computing the efficient communication distance and scheduling data transmission, where data transmission scheduling is separated into three separate algorithms for promoting efficient scheduling over short, medium, and long distances.

### 3.1. Efficient Communication Length Computation

Each sensor node in a cluster of the WSN has an associated efficient communication range that allows for the maximum transmission of ground data to the sink node of the UAV. Here, the efficient communication range in the flat of height *h* is characterized by a radius, *r_h_*, which is based on the known transmission power of the node and the altitude *h* of the UAV, which can be determined by the spatial resolution of the remote sensing image acquired according to the camera parameters of the UAV. However, the length of the UAV flight path through the efficient communication range of each sensor node will be different because the planned flight path of the UAV is independent of the sensor node locations. This length will determine the time period allowed for efficient communication between the UAV and a sensor node according to the known speed, *v*, of the UAV, and therefore determines the maximum quantity of data that can be efficiently transmitted from a sensor node to the UAV. Therefore, the selection of the communication node within a cluster must be conducted to maximize the quantity of data that can be efficiently transmitted. Accordingly, an efficient communication length algorithm was developed to compute the maximum amount of data that can be transmitted to the UAV by each sensor node in a cluster, based on the efficient communication length and *v*.

The DTORP uses turning points to segment a flight path into *m* straight lines. Hence, *m* = 1 for a perfectly straight flight path. For *n* sensor nodes on the ground, the perpendicular distance from each sensor node to each line segment *i* in the UAV flight path is denoted as *d_i_*_,*j*_, where *i* = 1, 2, …, *m* and *j* = 1, 2, …, *n*. It is easily determined that a line segment, *I*, cannot pass through the efficient communication area of sensor node *j* unless *d_i_*_,*j*_ < *r_h_*. Therefore, only line segments for which *d_i_*_,*j*_ < *r_h_* are considered for further analysis.

The six conditions illustrated in Figure 1 were considered for further analysis based on *r_h_* and the distances from sensor node *j* to the first and second turning points of line segment *i*, which are herein denoted as *L_i_*_−1,*j*_ and *L_i_*_,*j*_, respectively. Here, the six cases are divided according to whether the two turning points of a line segment are on different sides of the sensor node or on the same side and according to the values of *L_i_*_–1,*j*_ and *L_i_*_,*j*_, relative to *r_h_*. We also must note that the line segments in Figure 1 have all been rendered with horizontal orientations for simplicity, although these line segments can travel in any arbitrary direction. Defining the length of line segment *i* in the efficient communication range of sensor node *j* as *s_i_*_,*j*_, its value can be determined for the different and same side cases as follows: If both *L_i_*_−1,*j*_ and *L_i_*_,*j*_ are greater than *r_h_*, then *s_i_*_,*j*_ is defined as follows:(1)$$s_{i,j}=\{\begin{array}{c} 2\times \sqrt{r_{h}^{2}-d_{i,j}^{2}}\\ 0 \end{array}\;\;\;\begin{array}{c} Different\;side\\ Same\;side \end{array}.$$

If either *L_i_*_−1,*j*_ or *L_i_*_,*j*_ is greater than *r_h_* and the other is less than *r_h_*, then *s_i_*_,*j*_ is defined as follows:(2)$$s_{i,j}=\{\begin{array}{c} \sqrt{r_{h}^{2}-d_{i,j}^{2}}+\sqrt{\min{\left(L_{1},L_{2}\right)}^{2}-d_{i,j}^{2}}\\ \sqrt{r_{h}^{2}-d_{i,j}^{2}}-\sqrt{\min{\left(L_{1},L_{2}\right)}^{2}-d_{i,j}^{2}} \end{array}\;\;\;\begin{array}{c} Different\;side\\ Same\;side \end{array}$$

If both of *L_i_*_−1,*j*_ and *L_i_*_,*j*_ are less than *r_h_*, then *s_i_*_,*j*_ is defined as follows:(3)$$s_{i,j}=\{\begin{array}{c} \sqrt{L_{1}^{2}-d_{i,j}^{2}}+\sqrt{L_{2}^{2}-d_{i,j}^{2}}\\ \left|\sqrt{L_{1}^{2}-d_{i,j}^{2}}-\sqrt{L_{2}^{2}-d_{i,j}^{2}}\right| \end{array}\;\;\;\begin{array}{c} Different\;side\\ Same\;side \end{array}.$$

Finally, the maximum amount of data, *Q_j_*, that can be efficiently transmitted to the UAV by sensor node *j* can be calculated based on the data transmission rate, *T*, of the node as follows:(4)$$Q_{j}=\frac{T\times \sum_{i=1}^{m}s_{i,j}}{v}.$$

### 3.2. Data Transmission Scheduling

The data transmission scheduling adopted in the DTORP was developed to address differences between the maximum amount of data *Q* that can be efficiently transmitted to the UAV and the volume of data storage in each sensor node by adjusting the characteristics of data transmission within a sensor node cluster based on the neighboring node list of each sensor node. The algorithm seeks to obtain an optimal tradeoff between the volume of data transmission for each sensor node and the entire cluster, and the energy consumption of the WSN.

To this end, all sensor nodes in a cluster are divided into three node types, namely, either demand nodes, support nodes, or route nodes. A demand node has a value of *Q* that is less than its volume of data and must therefore transmit some of its data to other sensor nodes. A support node has a value of *Q* that is greater than its volume of data, and therefore has the capacity to receive data for demand nodes. Finally, route nodes have a value of *Q* that is equal to their volume of data and assist in transmitting data from demand nodes to support nodes when a demand node is unable to communicate directly with support nodes. The fundamental purpose of our algorithm is to facilitate the identification of suitable support nodes for enabling the necessary sharing of data storage resources among the available nodes in the WSN.

The proposed protocol further reduces energy consumption by combining the scheduling algorithm and routing algorithm employed for conducting data transmission between the nodes in a cluster over three distance scales, including short, medium, and long distances. These separate combined scheduling and routing algorithms are presented in the following subsections.

#### 3.2.1. Short Distance Scheduling: Unit Data Polling Scheduling and the Maximum Remaining Energy Routing Algorithm

The conditions of short distance scheduling occur when the demand nodes require only one jump or two for transmitting data to the support nodes. Here, single-jump transmission represents a condition where the demand nodes can transmit data to the support nodes directly, while two-jump transmission represents a condition where the demand nodes require another sensor node as a relay for transmitting data to the support nodes.

The Algorithm 1 adopts unit data polling scheduling, which functions similarly to the round robin scheduling employed in cloud computing technology. Here, one unit of data is assigned in each round of the scheduling process to the support node lying within the shortest distance from each demand node. The assignment process ends when all the data transmission tasks of the demand nodes have been assigned or all the support nodes have no remaining data requiring transmission. The routes from the demand nodes to the support nodes are selected to maximize the remaining energy of the relay nodes.

In this case, all demand nodes have the same probability of distributing their data to each support node lying within the shortest distance. The algorithm can maximize the volume of data transmission and make full use of the support nodes to reduce the energy consumption of the data transmission process between demand nodes and support nodes.
**Algorithm 1.** Unit Data Polling Scheduling and the Maximum Remaining Energy Routing Algorithm.1:Initialize: The complete list *N* of sensor nodes and the number of sensor nodes *n*, list *D* of demand nodes and the number of demand nodes *n_dn_*, list *S* of support nodes and the number of support nodes *n_sn_*, and the list of data sending *DS* and data receiving *DR* for all sensor nodes.2:While *n_dn_* > 0, *n_sn_* > 0 and support nodes exist within a one- or two-jump distance from the demand nodes, do3:  For each *D_i_*, *i* ∈ *n_dn_* and *S_j_, j* ∈ *n_sn_*, do4:     If *D_i_* and *S_j_* lie within a single jump distance, then5:        Record the route and accumulate *DS_i_* and *DR_j_*6:        Update the quantity of data transmitted for *D_i_* and *S_j_*7:     End if8:    If *D_i_* and *S_j_* lie within a two-jump distance, then9:      Find all routes from *D_i_* to *S_j_,* and select the route *N_k_* that has the greatest remaining energy10:      Record the route and accumulate *DS_i_*, *DR_j_*, *DS_k_*, and *DR_k_*11:      Update the quantity of data transmitted for *D_i_* and *S_j_*12:    End if13:  End for14:End while15:Compute the energy consumption using *DS* and *DR*.16:Update the remaining energy for all nodes.

#### 3.2.2. Medium Distance Scheduling: Maximum Data Greedy Scheduling and the Maximum Remaining Energy Routing Algorithm

The conditions of medium distance scheduling occur when the demand nodes require three or four jumps for transmitting data to the support nodes. As such, the data routes pass through two or three different sensor nodes which serve as route nodes.

The Algorithm 2 adopts maximum data greedy scheduling, which is a dynamic allocation algorithm that is similar to the greedy algorithm. Here, the demand node having the highest data transmission load assigns its data to the supply node within a distance of three or four jumps that has the greatest additional data transmission capacity. The list of demand nodes is then reorganized after each round of assignment according to the quantity of transmitted data from largest to smallest. As was adopted for short distance scheduling, route planning is conducted over medium distances using the maximum remaining energy routing algorithm as well.

The primary advantages of this algorithm are that it reduces the number of assignments to reduce energy consumption during data transmission and that it balances the remaining energy of nodes over the cluster.
**Algorithm 2.** Maximum Data Greedy Scheduling and the Maximum Remaining Energy Routing Algorithm.1:Initialize: The complete list *N* of sensor nodes and the number of sensor nodes *n*, the list *D* of demand nodes, the quantity of data *DD* that must be transmitted, and the number of demand nodes *n_dn_*, list *S* of support nodes, the quantity of data *SD* available for support, and the number of support nodes *n_sn_*. 2:Sort *D* according to *DD* and *S* according to *SD* in descending order3:If *n_dn_* > 0, *n_sn_* > 0 and support nodes exist within a three- or four-jump distance, then4:  For each *D_i_*, *I* ∈ *n_dn_* and *S_j_, j* ∈ *n_sn_*, do5:    If *D_i_* and *S_j_* lie within three or four jumps distance, then6:      If *DD_i_* ≥ *SD_j_* then7:        *DD_i_* = *DD_i_* − *SD_j_*8:        *SD_j_* = 09:      Else10:        *DD_i_* = 011:        *SD_j_* = *SD_j_* − *DD_i_*12:       End if13:       Find all routes from *D_i_* to *S_j_*14:       Select the route having the greatest remaining energy15:       Record the route and the amount of data transmitted16:     End if17:  End for18:End if

#### 3.2.3. Long-Distance Scheduling: Maximum Data Greedy Scheduling and the Diffusion Route Finding Algorithm

The conditions of long-distance scheduling occur when the demand nodes require more than four route nodes for transmitting data to the support nodes. Under this condition, the demand nodes and support nodes can be located anywhere within the cluster.

The Algorithm 3 also adopts maximum data greedy scheduling, except that the distance between the demand node and the support node is neglected in the assignment process. It can reduce the computational time of the whole algorithm and reduce the number of routes required for conducting data transmission. Route planning is conducted over long distances using the diffusion route finding algorithm to obtain the minimal number of jumps required for data transmission from the demand nodes to the support nodes.

The primary advantages of this algorithm are that it requires a relatively short computational time, reduces the number of routes required for conducting data transmission, and also reduces the required number of jumps in each route. Thus, the energy consumption of the entire cluster can be reduced during data transmission over long distances.
**Algorithm 3**. Maximum Data Greedy Scheduling and the Diffusion Route Finding Algorithm.1:Initialize: The complete list *N* of sensor nodes and the number of sensor nodes *n*, the list *D* of demand nodes, the quantity of data *DD* that must be transmitted, and the number of demand nodes *n_dn_*, the list *S* of support nodes, the quantity of data transmission *SD* available for support, and the number of support nodes *n_sn_*. 2:While *n_dn_* > 0 and *n_sn_* > 0, do3:  Sort *D* according to *DD* and *S* according to *SD* in descending order4:  If *DD_l_* ≥ *SD_l_*, then5:    *DD_l_* = *DD_l_* − *SD_l_*6:    *SD_l_* = 07:    *n_sn_ = n_sn_* − 18:   Else9:    *DD_l_* = 010:    *SD_l_* = *SD_l_* − *DD_l_*11:    *n_dn_* = *n_dn_* − 112:  End if13:    While *S_l_* has not been found, then14:    For all nodes lying one more jump away from *D_l_*, do15:      If *S_l_* is found in these nodes, then16:        Record the route17:        Break18:      End if19:    End for20:  End while21:End while

## 4. Simulations

The performance of the improved UAV-WSN system and the DTORP proposed in this study was assessed for a WSN composed of a single cluster using Monte Carlo simulations for different spatial resolution images captured by a UAV based on comparisons with the performance obtained for a conventional UAV-WSN system employing the LEACH protocol.

### 4.1. Simulation Setup

We used the C# programming language and Visual Studio 2017 to program an emulator for the UAV-WSN system, using SQL Server 2017 to manage the emulator data.

The area of the simulated region was 500 × 500 m. The number of sensor nodes in the cluster was 10. The parameters of the sensor nodes and the sink node were based on those of the CC2530 (Texas Instruments, Inc., Dallas, TX, USA) second generation system-on-chip (SoC) solution for Low Rate Wireless Personal Area Network (LR-WPAN), Zigbee, and RF4CE applications, which is one of the most commonly employed communication modules in WSNs. To ensure accuracy, numerous node parameters were considered, such as the communication rate, sending current, and receiving current. Each sensor node was set to record four values as a single unit of data and save 12 units of data every day. All the heights of the nodes were set to be 0 m. The aircraft and camera parameters of the simulated UAV were based on the DJI Phantom 3 Advanced drone, which is a quadcopter with a complementary metal oxide semiconductor (CMOS) sensor for vertical photograph capture. The primary parameters include a maximum UAV speed of 16 m/s, a maximum flight time of about 23 min, and an image size of 4000 × 3000 pixels. The side overlap of the images captured by the drone was 60%. The speed of the drone was held at a constant 8 m/s and the flight height of the drone was held when in the planned flight path. The spatial resolutions of the images varied from 0.5 × 0.5 cm to 4 × 4 cm in increments of 0.5 cm. The altitude of the drone and the density of the flight lines were determined according to the spatial resolutions of the images and other parameters. We conducted 1000 simulations with the 10 sensor nodes of the WSN randomly deployed in the simulated region for each spatial image resolution considered. The sensor node deployment was constrained to ensure that every sensor node was able to communicate with at least one other sensor node in the WSN. In addition, the total coverage area of UAV imaging is always greater than the area of the farmland, and various flight paths can be adopted to effectively image the selected region. Therefore, 10 flight paths that appropriately imaged the farmland area were randomly configured within the simulations. Accordingly, 1000 × 10 simulations were conducted for each spatial resolution considered. Three performance indicators were adopted for evaluating system performance. These included the total volume of data transmitted by the ground cluster of the WSN and the sink node of the UAV, the energy efficiency, and the maximum revisit period. Here, the energy efficiency represents the efficiency of energy utilization for each system, including the energy used for sending and receiving data to and from ground sensor nodes and the sink node of the UAV, and was defined as the energy required for the transmission of one kB of data. Accordingly, the efficiency of energy utilization of a UAV-WSN system decreases as the energy efficiency value increases. The maximum revisit period was defined as the maximum time (days) allowed for the UAV-WSN system to re-visit the sensor nodes on the ground to collect a given amount of data. As such, this indicator refers to the frequency of revisits required to collect the data from the sensor nodes, and therefore reflects the long-term monitoring efficiency of the system. For a given amount of data, the frequency of revisits increases, and the allowed maximum revisit period decreases as the volume of data transmitted by the system decreases.

### 4.2. Performance of the Improved UAV-WSN System

The relationship between the total volume of data transmitted by the cluster and the spatial resolution of UAV images is given in Figure 2. We note that the total volume of the transmitted data decreased as the spatial resolution of the UAV images became increasingly coarse for both systems. This is because the altitude of the UAV and the distance between the neighboring flight paths increases with increasing pixel size. This increases the distance between the sink node in the UAV and the sensor nodes on the ground, which reduces the area of the effective communication range of the ground sensors, and thereby decreases the time available for effective communication between the sink node and the sensor nodes.

These results demonstrate that the total volume of data transmitted in the improved UAV-WSN system using the DTORP was much greater than the existing UAV-WSN system using the LEACH protocol. Here, the improved UAV-WSN system collected more data from the ground sensor node cluster for each flight mission than the existing UAV-WSN system, specifically, by a factor of seven, irrespective of the spatial image resolution. This is because the DTORP in the improved UAV-WSN system considers every sensor node in the WSN as a potential communication node that can transmit data to the sink node of the UAV. Thus, the DTORP increases the maximum quantity of data that can be transmitted by the nodes, because the sink node can receive data whenever the UAV flies over the communication range of the overall sensor node cluster. Meanwhile, the LEACH protocol in existing UAV-WSN systems uses only the head node in the cluster for data transmission. Therefore, the improved UAV-WSN system can greatly increase the data transmission efficiency of the existing UAV-WSN system.

The relationship between the energy efficiency of the UAV-WSN systems for data transmission and the spatial resolution of UAV images is given in Figure 3. We note that the energy efficiency values of the conventional UAV-WSN system employing the LEACH protocol indicate that the energy consumed by this system in data transmission was very stable. This is because the LEACH protocol adopts a uniform scheduling rule for transmitting data. However, its energy efficiency at each spatial image resolution was considerably less than that obtained for the improved UAV-WSN system employing the proposed DTORP. We further note that the energy consumption of the improved UAV-WSN system was also quite stable, at least until the image pixel size was greater than 2 × 2 cm, and that the energy consumption of the system generally increased with increasing pixel size thereafter. This is the result of applying three different scheduling protocols in the DTORP for small, medium, and large distances between demand nodes and support nodes. However, the lower energy consumption confirms the benefits of this process as the spatial image resolution becomes increasingly coarse.

These results demonstrate that the energy efficiency value of data transmission obtained using the DTORP was much smaller than that obtained using the LEACH protocol. Here, the DTORP consumed less than one-fifth of the energy consumed by the conventional LEACH protocol when the pixel size of the image was not more than 3.5 × 3.5 cm. Moreover, the energy consumption of data transmission obtained by the DTORP for a spatial image resolution of 4 × 4 cm remains two-fifths of that consumed by the LEACH protocol. The benefits of the three-step data scheduling process employed by the DTORP for reducing energy consumption during data transmission are particularly evident as the spatial image resolution becomes increasingly coarse and the data transmission load increases. Here, a coarse spatial image resolution is obtained when the UAV is at a high altitude, and a relatively large number of sensor nodes must transmit their data to other sensor nodes to accommodate the relatively small effective communication ranges involved. Accordingly, all the steps of the DTORP can be fully applied to optimize the data transmission process and reduce the corresponding energy consumption.

The relationship between the maximum revisit period of the UAV-WSN systems and the spatial resolution of UAV images is given in Figure 4. We note that the maximum revisit period of the proposed UAV-WSN system was, in fact, greater than one year for pixel sizes less than 2.5 × 2.5 cm, and it decreased continuously as the spatial resolution became increasingly coarse. Here, in the Figure 4, the max value of the y-axis was limited to 365 days to clearly illustrate this decreasing trend. The conventional UAV-WSN system exhibited a similar decreasing trend for the maximum revisit period, except with much smaller values than the proposed system, which is mainly the result of the much smaller total volume of transmitted data (Figure 2). This indicates that the conventional UAV-WSN system requires a much higher frequency of revisits to collect the data from the sensor nodes than the proposed system for a given study area and spatial image resolution.

Overall, the one-year maximum revisit period obtained by the improved system when the spatial image resolution is finer than 2 × 2 cm indicates that this consideration is not a significant issue when revisiting is scheduled for the improved UAV-WSN system.

The above results obtained for the three performance indicators demonstrate that the DTORP can greatly improve the data transmission efficiency and reduce the energy consumption of the improved UAV-WSN system relative to those of a conventional UAV-WSN system employing the widely used LEACH protocol. These results, when taken together, demonstrate that the improved UAV-WSN system can be expected have a longer lifetime and be adaptive for long-term monitoring applications.

### 4.3. Discussion

The improved UAV-WSN system employing the DTORP proposed in this study has demonstrated its potential for using sensor nodes on the ground to monitor changes in the environment directly at sampled locations while simultaneously using drones to monitor the entire region. As such, in addition to farmland quality monitoring, the proposed system is equally applicable to many different activities that seek to utilize UAV-based remote sensing images and WSN-based ground sensing for monitoring data simultaneously, such as environmental monitoring, pollution control, flood condition monitoring, and emergency response. The combination of the time-varying characteristics obtained from long-term WSN sensing data and the spatially-varying characteristics obtained from high spatial resolution remote sensing images can lead to the development of accurate spatiotemporal models for the variables of interest in these applications. 

## 5. Conclusions

This study addressed the shortcomings of conventional UAV-WSN systems for conducting long-term farmland quality monitoring by designing a monitoring system that facilitates the integrated and simultaneous collection of high-resolution remote sensing and long-term ground sensing data based on a newly proposed DTORP. In contrast to existing UAV-WSN systems that plan UAV flight paths according to the locations of communication nodes in the WSN, the proposed system conducts flight planning independently of the node positions by considering every sensor node in a cluster to be a potential communication node that can transmit data to the sink node of the UAV. The proposed DTORP selects the communication node within a cluster of the WSN to maximize the quantity of data that can be efficiently transmitted based on an efficient communication length algorithm. In addition, the proposed protocol combines the scheduling algorithm and routing algorithm employed for conducting data transmission between the nodes in a cluster over three distance scales, including small, medium, and long distances, in order to reduce the energy consumption incurred during the data transmission process. The performance of the improved UAV-WSN system employing the DTORP was assessed for a WSN composed of a single cluster using Monte Carlo simulations for different spatial resolution images captured by a UAV, based on comparisons with the performance obtained for a conventional UAV-WSN system employing the LEACH protocol. The simulation results demonstrated that (1) the proposed system had better overall performance for all three performance indicators than the conventional system, (2) that the DTORP facilitated the transmission of a much greater total volume of data than the LEACH protocol, (3) that the DTORP utilized sensor node energy for data transmission with much greater efficiency than the LEACH protocol, (4) and that the DTORP provided a much larger maximum revisit period than the LEACH protocol. While the proposed UAV-WSN system has achieved several improvements compared with conventional UAV-WSN systems, further studies should be conducted. In the future, we will conduct more research on improved UAV-WSN systems based on DTORP to increase the volume of transmitted data and reduce energy consumption and apply it for the monitoring of the dynamics of farmland quality using a mobile farmland quality monitoring laboratory. Future efforts could be directed in various directions. Firstly, a sleep strategy for sensor node clusters could be introduced to further reduce energy consumption. Secondly, a redundant backup or storage algorithm must be introduced, because sensor nodes may breakdown and data may be lost prior to revisitation. Finally, we note that actual UAV flight paths do not always conform to the straight lines formulated in the planning stage because of numerous factors, such as strong winds and errors in the GPS coordinates of the drone. Also, the packet losses of the WSN are not 0%, because of the influence of various factors such as vegetation and buildings. All these influencing factors are regarded as uncertainties that should be considered. To this end, an adaptive UAV flight speed adjustment method should be developed to ensure that the UAV can collect all pertinent data under any condition encountered during applications. Furthermore, we will investigate the benefits of adopting different types of UAVs and different communication methods, such as Wi-Fi and Long Range (LoRa) in different applications.

## Figures and Tables

**Figure 1 sensors-20-00336-f001:**
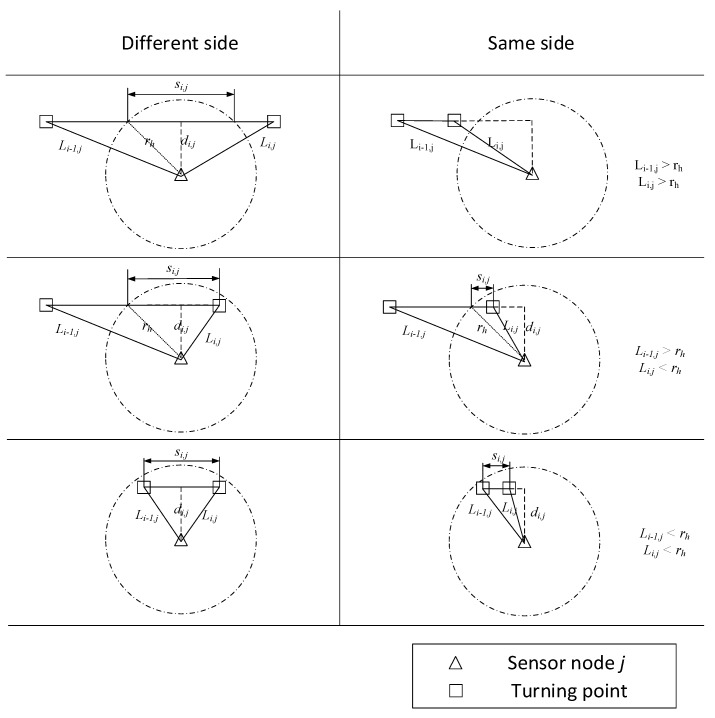
Six conditions where flight line segment *i* may pass through the efficient communication range of sensor node *j*, which depend on the efficient communication radius *r_h_* and the distances from the sensor node to the first and second turning points of the line segment (*L_i_*_−1,*j*_ and *L_i_*_,*j*_).

**Figure 2 sensors-20-00336-f002:**
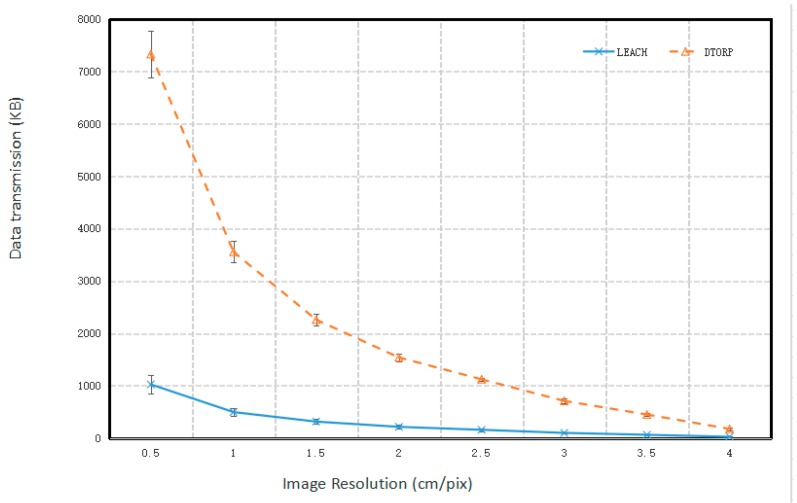
Total volume of data transmitted by the 10 sensor nodes in the wireless sensor network (WSN) clusters, with respect to the spatial image resolution of the unmanned aerial vehicle (UAV). LEACH: Low-energy adaptive clustering hierarchy. DTORP: Data transmission optimization routing protocol.

**Figure 3 sensors-20-00336-f003:**
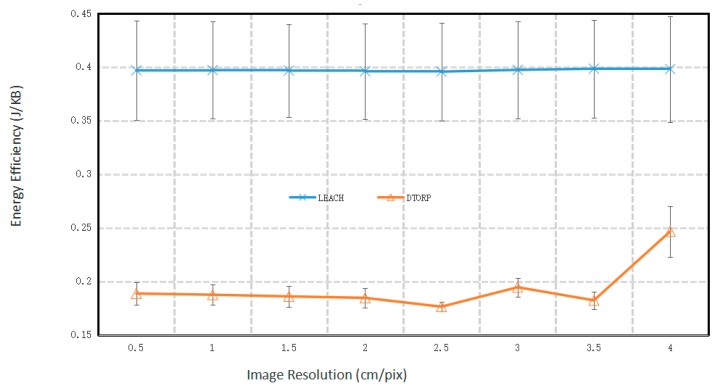
Energy efficiency of the UAV-WSN systems with respect to the spatial image resolution.

**Figure 4 sensors-20-00336-f004:**
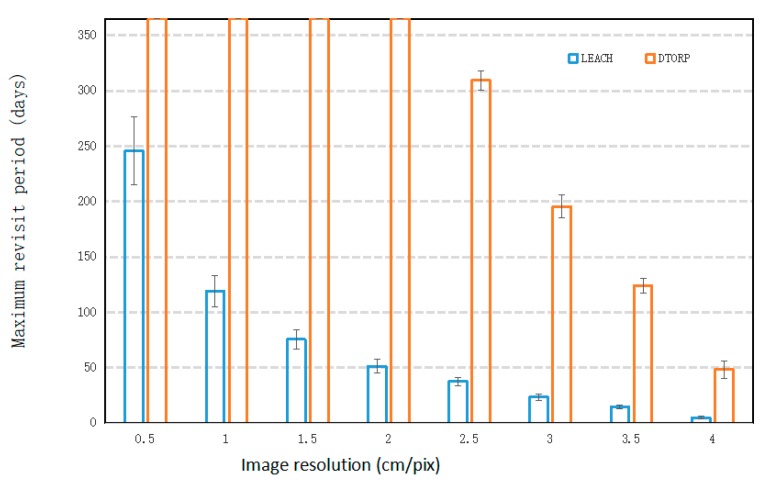
Maximum revisit period of the UAV-WSN systems with respect to the spatial image resolution.

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
