# Peer review of "Design of an Integrated Remote and Ground Sensing Monitor System for Assessing Farmland Quality"

_sensors, 2020, doi:10.3390/s20020336_

Round 1
Reviewer 1 Report
Please, see the attached file.

Reviewer 2 Report
The authors propose a new data transmission method optimized to gather as much as possible data from ground stations without changing a UAV flight plan.
In the introductory part, the authors could inform a little bit more on the type of data, and/or variety of data the system possibly works. This would link nicely with the chosen burden of 4 values saved 12 a day or even serve as an adjustment on this chosen value.
When presenting the algorithm, the complexity or processing time would be a nice addition. As the UAV flies by the ground base, if it calculates online, it is extremely important to measure this time and to account it on the maximum data transmitted. However, if the proposed algorithm is done a priori of UAV flight, then, the authors must inform it.
When explaining about the methodology, it would be nice to include in the commentaries the range of CC2530, alongside pixel resolution, the authors should include height of measurement; alongside the velocity, the authors should state our calculate how much of overlap between 2 frames, thus, it is possible to assess if the constant velocity chosen, can give the necessary overlap to build the map.
About the maximum revisit period as a metric, it is correlated with amount of data and how much it can be transmitted. Therefore, as a metric it has a correlation with the metric Data Transmission, not being a good choice for comparisons.
As the DTORP has 3 different algorithms, depending on distance, it would be nice an addition of metrics such as Energy or Data transmission, compared with the 3D distance. As they have 80000 simulations, they could, in simulation store the distance value, when the communication happened and, for presenting data, they could use few intervals to ease readers, for instance a 10-cm resolution.
Finally, the authors claim that the proposed methodology ease integrated and simultaneous collection of data. An important point, that the authors may want to discuss, perhaps in introduction, is that the simulations were about a system that have, in ground, 12 log on a day, and thus, they have specific timestamps, compared to the image collected on flight, in a different timestamp. Thus, the authors propose to solve how to collect data from the stations on ground without the need to approach the UAV. However, if, as a premise for the simulations, the synchronization is not intended, does a new routing method, for the WSN, to gather in a specific point, not a better solution in filed? That way, UAV path is still modular, WSN and UAV data have different timestamps, anyway, and the UAV would not need to carry extra load.
Reviewer 3 Report
The paper presents an interesting premise of work for efficient data collection using UAV-WSN system. A novel approach for data transmission is discussed based on communication length computation and data scheduling over short, medium and long distance.
While the approach is explained clearly, the proposed approach does not comment on the following
1) Energy awareness of the scheduling - how is the energy balanced across the network if the same support and route nodes are selected over time.
2) What is the overhead cost of setting up the scheduling? How often is this performed and the effect it has on the energy cost and bandwidth, especially in a dynamic topology.
For the evaluation section, the performance has been evaluated in terms of data transmission, energy efficiency and maximum revisit period against the image resolution. It is not clear what image resolution is desirable to have useful data from the UAVs? Also, the evaluation presented is trivial and is performed on a set of 10 nodes. It would be interesting to see the performance of the algorithm for a large network given simulation is used. I believe the effect of number and density of nodes must be evaluated for performance analysis. Furthermore, effect of change in DD values should be shown to understand the impact on performance.
